# PREVENTING IMITATION LEARNING
# WITH ADVERSARIAL POLICY ENSEMBLES

## ABSTRACT

Imitation learning can reproduce policies by observing experts, which poses a problem regarding policy privacy. Policies, such as human, or policies on deployed robots, can all be cloned without consent from the owners. How can we protect our proprietary policies from cloning by an external observer? To answer this question we introduce a new reinforcement learning framework, where we train an ensemble of near-optimal policies, whose demonstrations are guaranteed to be useless for an external observer. We formulate this idea by a constrained optimization problem, where the objective is to improve proprietary policies, and at the same time deteriorate the virtual policy of an eventual external observer. We design a tractable algorithm to solve this new optimization problem by modifying the standard policy gradient algorithm. Our formulation can be interpreted in lenses of confidentiality and adversarial behaviour, which enables a broader perspective of this work. We demonstrate the existence of "non-clonable" ensembles, providing a solution to the above optimization problem, which is calculated by our modified policy gradient algorithm. To our knowledge, this is the first work regarding the protection of policies in Reinforcement Learning.

## 1 INTRODUCTION

Imitation learning and behavioral cloning provide really strong ability to create powerful policies, as seen in robotic tasks (Laskey et al., 2017; Finn et al., 2017; Codevilla et al., 2019; 2017; Pomerleau, 1988; Bojarski et al., 2016). Other fields have developed methods to ensure privacy (Al-Rubaie & Chang, 2019; Papernot et al., 2016), however, such work do not offer *protection* against policy cloning.

In this work, we tackle the issue of protecting policies by training policies that aim to prevent an external observer from using behaviour cloning. Our approach draws inspiration from imitating human experts, who can near-optimally accomplish given tasks. The setting which we analyze is presented in Figure 1. We wish to find a collection of experts, which as an ensemble can perform a given task well, however, also targets behaviour cloning through adversarial behaviour. Another interpretation is that this collection of experts represents the worst case scenario for behaviour cloning on how to perform a task "good enough".

Imitation learning frameworks generally make certain assumptions of the optimality of the demonstrations (Ziebart et al., 2008; Levine, 2018), yet never considered the scenario when the experts specifically attempt to be adversarial to the imitator. We pose the novel question regarding this assumption: does there exist a set of experts that are adversarial to an external observer trying to behaviour clone?

We propose Adversarial Policy Ensembles (APE), a method that simultaneously optimizes the performance of the ensemble and minimizes the performance of policies eventually obtained from cloning it. Our experiments show that APE do not suffer much performance loss from an optimal policy, while causing, on average, the cloned policy to experience over 5 times degradation compared to the optimal policy.

Our main contributions can be summarized as follows:

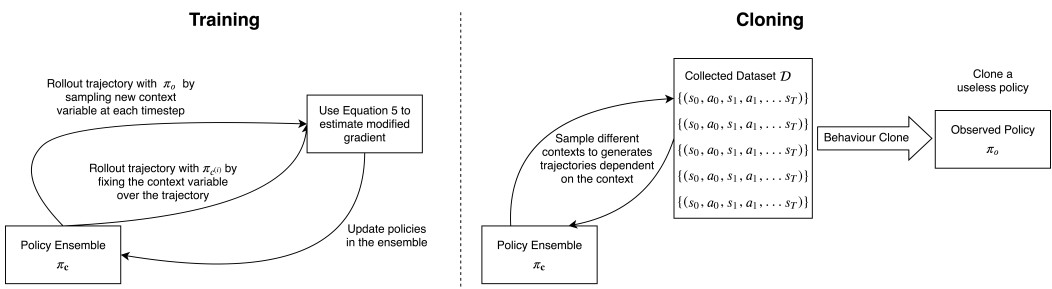

Figure 1: **Confidentiality scheme**: **Left** During training, optimize a Policy Ensemble by estimating gradients using both the policies in the ensemble and the fictitious observer policy. **Right** When collecting a dataset for cloning, the context variable is marginalized out. Thus cloning the Policy Ensemble can result in a useless policy

- We introduce a novel method APE, as well as the mathematical justification of the notion of adversarial experts.
- By modifying Policy Gradient (Sutton et al., 2000), a common reinforcement learning algorithm, we suggest a tractable scheme for finding an optimal solution for this objective.
- We demonstrate the solution by numerical simulations, where we show that a cloned policy is crippled even after collecting a significantly large number of samples from a policy ensemble.

To our knowledge, not only is this the first work regarding the protection of policies in reinforcement learning, but it is also the first to represent adversarial experts.

## 2 PRELIMINARIES

We develop APE in the standard framework of Reinforcement Learning (RL). The main components we use are Markov Decision Processes, Policy Gradient (Sutton et al., 2000), policy ensembles, and behaviour cloning, which we review below.

### 2.1 MARKOV DECISION PROCESS

A discrete-time finite-horizon discounted Markov decision process (MDP) $\mathcal{M}$ is defined by $(\mathcal{S}, \mathcal{A}, r, p, p_0, \gamma, T)$ where $\mathcal{S}$ is the state space, $\mathcal{A}$ is the action space, $r : \mathcal{S} \times \mathcal{A} \to \mathbb{R}$ is the reward function, $p(s_{t+1}|s_t, a_t)$ is the transition probability distribution, $p_0 : \mathcal{S} \to \mathbb{R}^+$ is the initial state distribution, $\gamma \in (0, 1)$ is the discount factor, and $T$ is the time horizon. A trajectory $\tau \sim \rho_\pi$, sampled from $p$ and a policy $\pi : \mathcal{S} \times \mathcal{A} \to \mathbb{R}^+$, is defined to be the states and actions tuple $(s_0, a_0, ...s_{T-1}, a_{T-1}, s_T)$, whose distribution is characterized by $\rho_\pi$. Define the return of a trajectory to be $r(\tau) = \sum_{t=0}^{T-1} \gamma^t r(s_t, a_t)$ to be the sum of discounted rewards seen along the trajectory, and define a value function $V^\pi : \mathcal{S} \to \mathbb{R}$ to be expected return of a trajectory starting from state $s$, under the policy $\pi$. The goal of reinforcement learning is to find a policy that maximizes the expected return $\mathbb{E}_{\tau \sim \rho_\pi}[r(\tau)]$.

### 2.2 POLICY GRADIENT

Policy Gradient (PG) (Sutton et al., 2000) aim to directly learn the optimal policy $\pi$, parameterized by $\theta$, by repeatedly estimating the gradient of the expected return, in one of many forms, shown in Schulman et al. (2015). In our work, we follow notation similar to that of (Schulman et al., 2015; 2017) and estimate $\nabla_\theta \mathbb{E}_{\tau \sim \rho_\pi}[r(\tau)]$ using the advantage, which is estimated from a trajectory $\tau$, $A_\tau^\pi(t) = R_\tau(t) - V^\pi(s_t)$, where $R_\tau(t) = \sum_{t'=t}^{T-1} \gamma^{t'} r(s_{t'}, a_{t'})$ is the sum of the reward following action $a_t$.

Here, the value function is learned simultaneously with the the policy, and so the advantage will use $\hat{V}^\pi$ as an estimate for $V^\pi$.

## 2.3 POLICY ENSEMBLE (PE)

A Policy Ensemble (PE) is similar to the notion of contexts and skills (Achiam et al., 2018; Eysenbach et al., 2018; Sharma et al., 2019) which we discuss in Section 3. We denote a PE by $\pi_c$, where each $\pi_{c^{(i)}}, i \in \{1, 2, ...n\}$ represents an expert. To rollout the PE, an expert is chosen at random (in our case uniform), and the expert completes a trajectory. Each expert policy $\pi_{c^{(i)}}(a|s)$ can be viewed as a policy conditioned on a latent variable $c$, $\pi(a|s, c)$.

Although $\pi_c$ consists of multiple policies, it is important to note that it itself is still a policy.

## 2.4 BEHAVIOUR CLONING

To behaviour clone an expert policy (Widrow & W. Smith, 1964), a dataset of trajectories $\mathcal{D}$ consisting of state action pairs $(s, a)$ are collected from the the expert rollouts. Then, a policy parametized by $\phi$ is trained to maximize the likelihood of an action given a state, $\sum_{(s,a)\in\mathcal{D}} -\log \pi_\phi(a \mid s)$.

When cloning $\pi_c$, $\mathcal{D}$ will not contain information of the latent variable $c$, and so the *cloned* policy will marginalize it out. Thus, the observer will clone:

$$\pi_o(a \mid s) := \sum_i p(c^{(i)} \mid s)\pi_{c^{(i)}}(a \mid s) \tag{1}$$

We stress that this policy does not exist until $\pi_c$ is behaviour cloned. $\pi_o$ is a fictitious policy to represent what would happen in the *best* case scenario of the observer having access to *infinite* data from $\pi_c$ to clone into $\pi_o$.

The scope of this paper is to specifically prevent behavioral cloning from succeeding. Other imitation learning approaches such as inverse reinforcement learning (Abbeel & Ng, 2004; Ng & Russell, 2000; Levine et al., 2011) and adversarial imitation learning (Ho & Ermon, 2016; Peng et al., 2018) require rollouts of non-expert policies in the environment, which may be costly, and thus are not considered.

## 3 RELATED WORK

**Adversarial Attacks in RL:** Our notion of adversarial policies is inextricably related to other adversarial methods that target RL such as Lin et al. (2017); Behzadan & Munir (2017), that add adversarial perturbations to policy input. Other adversarial attacks include poisoning the batch of data used when training RL (Ma et al., 2019), and exploitation in the multi-agent setting (Gleave et al., 2019). However, these methods all present as active attacks for various learning techniques. Our method, instead, passively protects against cloning.

**Privacy in RL:** With regards to protection, our work is related to differential privacy (Al-Rubaie & Chang, 2019). Differential privacy in RL can be used to create private Q-functions (Wang & Hegde, 2019) or private policies (Balle et al., 2016), which have private reward functions or private policy evaluation. However, we would like to emphasize that our motivation is to prevent cloning, and thus protecting the policies, rather than protecting a dataset. In fact, we make the assumption that the observer can perform behaviour cloning on as much data as desired.

**Imitation Learning:** Since we comply to the standard imitation learning setting of cloning from a dataset with many experts providing the demonstrations, latent variables w.r.t. imitation learning is well-studied. For example, Codevilla et al. (2017) show that conditioning on context representation can make imitation learning a viable option for autonomous driving. Li et al. (2017) demonstrate that the latent contextual information in expert trajectories is often semantically meaningful. As well, Providing extra context variables to condition on also appears in forms of extra queries or providing labels (Brown et al., 2019; de Haan et al., 2019; Hristov et al., 2018). Our method is different as instead of experimenting for success in imitation learning, we study how to prevent it.

**Multiple Policies:** (Achiam et al., 2018; Eysenbach et al., 2018; Sharma et al., 2019) have similar schemes of sampling a latent variable and fixing it throughout a trajectory, although their latent variables (contexts or skills) are used to solve semantically different tasks. The reason to solve *different* tasks is due to the objective of using the context variable/skills for learning in an unsupervised setting.

Our approach differs in both motivation and implementation, as we learn experts that all solve the same task, and constrain so that observers can not clone the policy.

A PE $\pi_c$ can also be viewed as a mixture of experts (Jacobs et al., 1991), except the gating network assigns probability 1 to the same expert for an entire trajectory. As such, we do not learn the gating network, although it may still be useful to see $\pi_c$ as a special case of a mixture of experts where the gating network learns immediately to fix the expert for each trajectory. There are also methods such as OptionGAN (Henderson et al., 2018), which uses a mixture of experts model to learn multiple policies as options with access to only expert states.

Zhang et al. (2019) also proposes a method to train multiple policies that complete the same task but uses the uncertainty of an autoencoder as a reward augment. Their motivation is to find multiple novel policies, while our motivation has no connection to novelty. Due to these differences in motivation, they train each policy one after the other, while our policies are trained simultaneously.

Policy ensembles are also used in the multi-task and goal conditioned settings in which case the task that is meant to be solved can be viewed as the context. Marginalizing out the context variable (Equation 1) of these context-conditioned policies is studied in the case of introducing a KL divergence regularizing term for learning new tasks (Goyal et al., 2019) and for sharing/hiding goals (Strouse et al., 2018). However, the main motivation is different in that both Goyal et al. (2019); Strouse et al. (2018) use $\pi_o$ to optimize mutual information, while we directly optimize its performance.

## 4 METHOD

### 4.1 OBJECTIVE

We wish to have experts that can perform the task, while minimizing the possible returns of the cloned policy, denoted in Equation 1. We modify the standard RL objective to be:

$$\arg\min_{\theta} \mathbb{E}_{\tau \sim \rho_{\pi_o}}[r(\tau)] \quad \text{s.t.} \ \mathbb{E}_{\tau \sim \rho_{\pi_c}}[r(\tau)] \geq \alpha \tag{2}$$

where $\alpha$ is a parameter that lower bounds the reward of the policy ensemble. This translates to maximizing the unconstrained Lagrangian:

$$J(\theta) = \mathbb{E}_{\tau \sim \rho_{\pi_c}}[r(\tau)] - \beta \mathbb{E}_{\tau \sim \rho_{\pi_o}}[r(\tau)] \tag{3}$$

where $1/\beta$ is the corresponding Lagrangian multiplier, and is subsumed into the returns collected by the policy ensemble. We refer to PE that optimizes this objective as Adversarial Policy Ensembles (APE). There is a natural interpretation of the objective in Equation 2. Human experts tend to be "good enough", which is reflected in the constraint. The minimization is simply finding the most adversarial experts.

Although we assume that the observer can only map states to actions, it may be the case that they can train a sequential policy, which is dependent on its previous states and actions. Our method can be generalized to sequential policies as well, and the impact of such observers is discussed in the Section 6.

### 4.2 MODIFIED POLICY GRADIENT ALGORITHM

Intuitively, since there are the returns of two policies that are being optimized, both should be sampled from to estimate the returns.

We show how we can modify PG to train APE, by maximizing Equation 3. The two terms suggest a simple scheme to estimate the returns of the policy ensemble twice: once using $\pi_c$ that we wish to maximize, and a second time using $\pi_o$, which approximates the returns of an eventual observer who tries to clone the policy ensemble. Along with our PE, we train value functions $\tilde{V}^{\pi_c(i)}$ for each expert, jointly parameterized by $\phi$ which estimates $V^{\pi_c(i)} - \beta V^{\pi_o}$. The loss function for the value functions of two sampled trajectories $\tau_1, \tau_2$ is

$$J_{\tau_1,\tau_2}(\phi) = \sum_{t=0}^{T_1-1} \frac{1}{2} \left( \tilde{V}_\phi^{\pi_{c^{(i)}}}(s_{t_1}) - R_{\tau_1}(t) \right)^2 + \sum_{t=0}^{T_2-1} \frac{1}{2} \left( \tilde{V}_\phi^{\pi_{c^{(i)}}}(s_{t_2}) + \beta R_{\tau_2}(t) \right)^2 \tag{4}$$

The policy gradient update from $N_1$ and $N_2$ trajectories is then

$$\nabla_\theta J_{\tau_1,\tau_2}(\theta) \approx G_1 + G_2 \tag{5}$$

where

$$G_1 = \frac{1}{N_1} \sum_{j=1}^{N_1} \sum_{t=0}^{T_1} \nabla_\theta \log \pi_{c^{(i)}}(a_{t1}^{(j)} \mid s_{t1}^{(j)}) \tilde{A}_{\tau_1}^{\pi_{c^{(i)}}}(t) \tag{6}$$

$$G_2 = \frac{1}{N_2} \sum_{j=1}^{N_2} \sum_{t=0}^{T_2} \nabla_\theta \log \pi_o(a_{t2}^{(j)} \mid s_{t2}^{(j)}) \tilde{A}_{\tau_2}^{\pi_o}(t) \tag{7}$$

where $c^{(i)}$ identifies the chosen expert of the trajectory., and $\tilde{A}_{\tau_1}^{\pi_{c^{(i)}}}(t) = R_{\tau_1}(t) - \tilde{V}^{\pi_{c^{(i)}}}(s_t)$ and $\tilde{A}_{\tau_2}^{\pi_o}(t) = -\beta R_{\tau_2}(t) - \tilde{V}^{\pi_o}(s_t)$ are the modified advantage functions. The $-\beta$ that is in the advantage in $G_2$ optimizes *against* the performance of the observed policy $\pi_o$.

The gradient $G_1$ for $\pi_{\mathbf{c}}$ is straightforward. However, to estimate the gradient $G_2$ for $\pi_o$ which is an fictitious policy, we sample from it by first re-sampling the context of the expert at each state, and then sampling an action from the context. The back-propagation occurs to $\pi_{c^{(i)}}(a \mid s)$ for the context sampled at each state. Practical implementation details can be found in A.2. The intuition is as follow. While sampling $\pi_o$, if a selected action causes *high* return, we should *decrease* the probability, which lowers the expected reward of $\pi_o$. Combined, the two gradients will cause the PE to select actions that achieves have high reward, and are detrimental to the observer.

Equations 4 and 5 formulate our PG approach of APE, which is summarized in Algorithm 1.

---

**Algorithm 1:** PG-APE

---

**Require:** $\theta$, $\phi$, $\mathcal{M}$, $\beta$
1: **for** each iteration **do**:
2:     Generate trajectories $\tau_1$ with $\pi_{\mathbf{c}}$ from $\mathcal{M}$ for Equation 6
3:     Generate trajectories $\tau_2$ with $\pi_o$ from $\mathcal{M}$ for Equation 7
4:     Calculate Equation 5 to perform a gradient update on the PE $\theta \leftarrow \theta + \alpha_\theta \hat{\nabla}_\theta J_{\tau_1,\tau_2}(\theta)$
5:     Update the value function $\phi \leftarrow \phi - \alpha_\phi \hat{\nabla}_\phi J_{\tau_1,\tau_2}(\phi)$ as determined by Equation 4.
6: **end for**

---

## 5 EXPERIMENTS

We perform experiments on a navigation task, where the objective is to reach a goal state as fast as possible. The purpose is to illustrate that an APE can cause the cloned policy to take significantly longer to reach the goal state. We do so by first training a PE and behaviour cloning it. We then compare the performance of the PE to that of the clone. We use a discrete environment to best demonstrate the validity of the equation. This is because all discrete policies can be parameterized, which is not true in continuous, where typically Gaussian parameterization is used. As such, continuous environments would have to make assumptions about how both the PE and the cloner parameterizes policies, as well as tackle problems of distributional drift, which we would like to avoid. However, with these assumptions, our setting can extend to the continuous domain. In our experiments, we use a $10 \times 10$ grid-world environment as our main testbed. This is to have large enough expression that would not be found in smaller grids, while still small enough to visualize the behaviour of the APE. The discrete actions will show precisely how the experts can be jointly adversarial.

Using gridworld allows for precise expected return estimates. In an environment where closed-form returns cannot be calculated, approximation error can accumulate through estimating the returns of

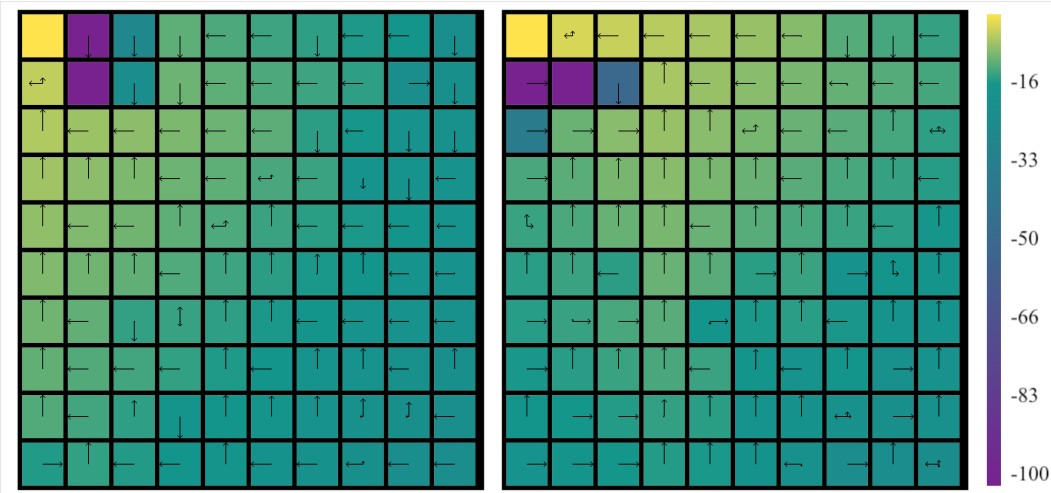

Figure 2: **Visualization of APE**. We set $\beta = 0.6$. Arrows indicate action probabilities, and the colour scale represents the hitting time. Yellow indicates expected reward of 0, while purple indicates expected reward of $-100$, which is the maximum episode length. The top left corner is the goal state, and the adjacent states that are purple are an example of how APE is adversarial to cloning, as those states will cause the cloned policy to suffer larger losses.

both the trained PE and the clone. This noise would only increase in continuous state space, where the returns of $\pi_o$ may not be tractable to estimate due to issues such as distributional drift (Ross et al., 2010; Codevilla et al., 2019; de Haan et al., 2019).

Our results answer the following questions. How much optimality is compromised? How useless can we make the cloned policy? Is it possible to use non APE to prevent behaviour cloning?

### 5.1 TRAINING

Even though our method can compute a policy ensemble with any finite number of experts, we chose to visualize a solution with 2 experts, which is sufficient to reveal the essential properties of the method. Specifically, we train $n = 2$ tabular experts with PG-APE. Our code is written in Tensorflow (Abadi et al., 2016), and will be publicly available on GitHub. Training details and hyper-parameters are in Section A.1 of the Appendix.

### 5.2 ENVIRONMENT

The basic environment is a $10 \times 10$ grid, with the goal state at the top left corner. The agent spawns in a random non-goal state, and incurs a reward of $-1$ for each time-step until it reaches the goal. At the goal state, the agent no longer receives a loss and terminates the episode. The agent is allowed five actions, $\mathcal{A} = \{$ *Up, Down, Left, Right, Stay* $\}$. Moving into the wall is equivalent to executing a *Stay* action. We choose this reward function for the benefit of having a clear representation of the notion of "good enough", which is reflected in how long it takes to reach the goal state. Having such representation exemplifies how the APE can prevent an observer from cloning a good policy.

### 5.3 VISUALIZATION

Figure 2 shows an example of a PE that is trained for the basic gridworld environment. Figure 3 shows the corresponding cloned policy, as well as a comparison to an optimal policy. The colour scale represents the expected return of starting at a given state.

In the case of an optimal policy ($\beta = 0$), actions are taken to take the agent to the goal state as fast as possible. However, when $\beta > 0$, such a solution is no longer the optimum. Similar to $\beta = 0$, the experts would like to maximize the expected reward, and reach the goal state. However, to minimize the reward of the observed policy, the two expert policies must jointly learn to increase the number

of steps needed for $\pi_o$ to reach the goal state. The expert policies must use adversarial behaviour while reaching the goal state, such as taking intelligent detours or *Stay* in the same state, which are learned to hinder $\pi_o$ as much as possible. These learnt behaviours cause the cloned policy to take a drastically longer time to reach the goal. For example, note the two purple squares at the top-left near the goal, which indicates that the experts understand that they should not move to prevent the observer from attaining reward. Even though these sub-optimal decisions are made, on expectation, the experts are "not bad" and achieve an average of $-15.27$ reward.

## 5.4 BASELINES

We use behaviour cloning to clone our PG-APE trained policies. To support our claims of preventing even in the horizon of infinite data, we collect a million timesteps of the trained PE in the environment. Further details of behaviour cloning are in the appendix. Shown in Figure 3 is an optimal policy, and the resulting cloned policy from Section 5.1.

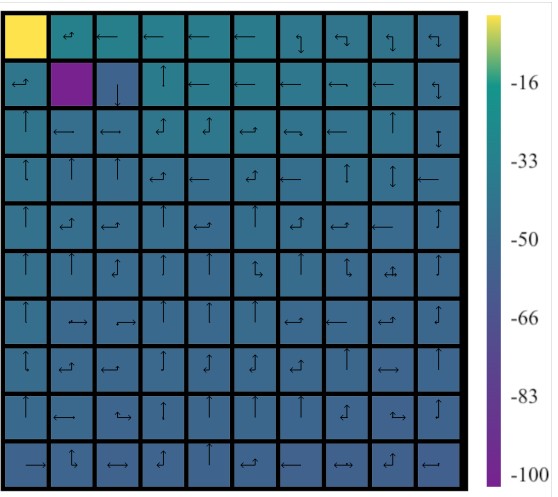

As well, we evaluate against other PE, to show that preventing against behaviour cloning is non-trivial. We use several baselines. We first test policies that have approximately the same return as our ensemble by training PG with, and halting early rather than running until convergence. In the Near-Optimal case, we ran until the PE had expected returns that matched the average achieved by our method. Conversely, "Random" policies are used as a comparison to show that it is possible to cause the cloned policy to do poorly, but the tradeoff is that the PE itself cannot perform well, which is undesirable. These policies are also policies trained with PG, except they are stopped much earlier, when their clones matches the expected returns of our PG-APE. For each PG-APE, we use $n = 2$ different tabular policies treated as an ensemble, which

Figure 3: **Visualization of the cloned APE**. The policy obtained from cloning the APE trained has average expected reward of $-45.18$, while the optimal policy has an average expected reward of $-9$, which is over a $5\times$ increase.

we then clone, and average across 5 seeds. For the baselines, we hand-pick the policies, and thus only use 3 different policies.

|  | PE Returns | Clone Returns | Returns Difference |
|---|---|---|---|
| PG-APE | -16.24 ±1.20 | -44.27 ±1.07 | **-28.03** |
| Near-Optimal PE | -16.74 ±1.32 | -16.67 ±1.31 | +0.07 |
| Random Policy | -44.59 ±0.52 | -44.52 ±0.77 | +0.07 |

Table 1: **Comparison of cloned PE**. Each policy has their Returns precisely calculated through their closed form solutions. The final column reports the difference between the PE and the Clone, which is only significant for our method.

As presented in Table 1, all other PE have an insignificant difference (returns of the PE subtracted from returns of the cloned policy) between the performance of the PE and the cloned policy, except for our method. These empirical findings show that preventing behaviour cloning difficult, but possible using APE.

## 6 DISCUSSION & FUTURE WORK

**Confidential Policies:** There are promising research directions regarding the protection of policies, due to the many applications where confidentiality is crucial. As long as there is a model of the observer, our presented method provides a worst-case scenario of experts.

In our work, we focused on the case where the observer does not use the current trajectory to determine their policy. Instead, it may be the case that the observer uses a sequential policy (one that depends on its previous states and/or actions), such as an RNN to determine the context of the current expert.

Formally, the observer will no longer learn the policy formulated in Equation 1 that is solely dependent on the current state, but rather a policy that is dependent on the current trajectory:

$$\pi_o(a \mid \tau_{1:t}) := \sum_i p(c^{(i)} \mid \tau_{1:t})\pi_{c^{(i)}}(a \mid s) \tag{8}$$

We found in our preliminary results that using an RNN classifier which outputs $p(c|\tau_{1:t})$ simply ended up in with either optimal policies or crippled policies. In both cases, there was a relatively minor difference in performance between the policy ensemble and the cloned policy.

Unsurprisingly, when the observer has access to a strong enough representation for their policy, then they should be able to imitate *any policy*. In this case, the worst-case set of experts cannot do much to prevent the cloning. We believe that this is an exciting conclusion, and is grounds for future work.

**Continuous:** Although our methods are evaluated in discrete state spaces, our approach can be generalized to continuous domains.

The Monte Carlo sampling in Equation 9 suggests that the use of continuous context may also be possible, given there is a strong enough function approximator to estimate the distribution of $c|s$. We see this as an exciting direction for future work, to recover the full spectrum of possible adversarial policies under the constraint of Equation 2.

**The Semantics of Reward:** Although the minimization in Equation 2 implies a logical equivalence between the success of behaviour cloning to the reward the cloned policy can achieve, it may follow that this is not the case. It may be the case that useless is defined differently by the expected reward the cloned policy achieves on a different reward function $\tilde{r}$. For example, a robot that is unpredictable should not be deployed with humans. Since the $r$ functions in Equation 2 are disentangled, the reward function $r$ that is minimized in Equation 2 can be engineered to fit any definition of uselessness.

We can modify the objective of APE by modifying Equations 4 and 5 to use a different reward function $\tilde{r}$ in the minimization, substituting $R(t)$ for $\tilde{R}(t) = \sum_{t'=t}^{T-1} \gamma^{t'-t}\tilde{r}(s_{t'}, a_{t'})$. The rest of the derivation and algorithm remain the same.

We think this is an exciting direction, especially for learning all different possible representations of the worst-case experts.

## 7 CONCLUSION

We present APE as well as its mathematical formulation, and show that policy gradient, a basic RL algorithm can be used to optimize a policy ensemble that cannot be cloned. We evaluated APE against baselines to show that adversarial behaviour is not feasible without our method.

This work identifies a novel yet crucial area in Reinforcement Learning, regarding the confidentiality of proprietary policies. The essence of our approach is that a policy ensemble can achieve high return for the policy owner, while providing an external observer with a guaranteed low reward, making proprietary ensemble useless to the observer.

The formulation of our problem setup and the algorithm are very general. In this first work we demonstrate the solution in the deliberately chosen simple environments in order to better visualize the essence of our method. In our concurrent work we study thoroughly the application of our method in various domains, which is out of the scope of this introductory paper.

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

# A  APPENDIX

## A.1  TRAINING DETAILS & HYPERPARAMETERS

For our training, we set $\alpha_\theta = 0.05$, and the value weight to be $0.5$, use annealed entropy regularization (Mnih et al., 2016) from $5e-1$ to $5e-3$ and set the discount factor $\gamma = 0.99$. Due to the contrasting gradients experienced, large batch sizes are used. In our experiments, we take 1 gradient update of AdaM (Kingma & Ba, 2015) per batch of 4096 (containing multiple trajectories), and trained for $3e6$ timesteps.

To estimate $p(c|s)$ in Equation 1, we use a replay buffer that keeps track of the previous 60 contexts seen at each state.

Estimating the quantity in Equation 8 requires memory, which we use a single GRU (Cho et al., 2014) as done in Strouse et al. (2018), with the exception that only states are fed in as a one-hot. Due to our environment is deterministic, state sequences captures the action sequence information. The single unit is then concatenated with the state, which feeds into a fully connected layer of 128, and then a soft-max, to produce the distribution $c|s$ over contexts.

For our behaviour cloning, we collect $1e6$ state action pairs, and train a tabular policy with $0.01$ learning rate on cross entropy softmax loss for 100 epochs. The large amount of data and epochs is to ensure that we can recover $\pi_o$ with little to no variance.

To solve the precise returns of the policies, we inject noise of $1e-9$, to ensure a hitting time always exists from each state. As well, we clip all the hitting times to $-T = -100$.

## A.2  ESTIMATING $\nabla_\theta \log \pi_o$

It is not obvious how $\nabla_\theta \log \pi_o$ *should* be estimated, since $\pi_o$ is never realized until the policy is cloned. Literally, it is a virtual policy.

Equation 1 offers a straightforward method to back-propagate, similar to that of the Mixture of Experts model Jacobs et al. (1991), except using an estimate of $c|s$ instead of a gating network.

However, we can also rewrite Equation 1 as $\sum_i p(c^{(i)}|s)\pi_{c^{(i)}}(a \mid s) = \mathbb{E}_{c \sim p(c|s)}[\pi_{c^{(i)}}(a \mid s)]$, which results in the gradient update being:

$$\nabla_\theta \log \pi_o(a|s) = \nabla_\theta \log \mathbb{E}_{c \sim p(c|s)}[\pi_{c^{(i)}}(a \mid s)] \tag{9}$$

which suggests a method of Monte Carlo sampling the inner expectation with 1 sampled context. Empirically, we use the Monte Carlo sampling method.

