# OpenReview forum: "Preventing Imitation Learning with Adversarial Policy Ensembles"
_ICLR.cc/2020/Conference — Reject_

### Official Review · AnonReviewer3 · 2019-10-23
**Official Blind Review #3**

**Rating:** 3

**Review:**

This paper addresses the problem of poisoning behavioral cloning using an optimized ensemble of demonstrators. The goals is allow the ensemble to still achieve an expected return above a certain threshold while minimizing the return of a policy trained via behavioral cloning.

This is a very exciting and novel paper, but it is not yet ready for publication. There are many typos and the paper is difficult to read at times. Also, the experiments are still very basic. While interesting, further experiments in more complicated discrete or continuous domains would greatly enhance the work.

I would recommend not focusing on the privacy of human policies. I think a better motivation is to focus on theoretical ideas of adversarial inputs to behavioral cloning to study robustness as well as potential counter-intelligence strategies for autonomous agents.

This work has similarities to machine teaching and poisoning attacks. It would be interesting to see if recent methods for machine teaching for IRL [1] or poisoning for RL [2] can be used to solve the proposed problem. It would be good to situate this work within these related works. It seems like the proposed problem can be seen as a kind of anti-machine teaching for IRL where the goal is to find a set of good demonstrations that are maximally uninformative.

Second paragraph in 2.3: It's unclear what is the point of this paragraph. I would recommend not focusing so much on human demos.

The min-max approach seems related to GANs and Generative Adversarial Imitation Learning. Can something similar be used to scale this approach to high-dimensional tasks?

Equations (4) and (5) are difficult to unpack. It would be nice add a little more explanation and intuition.

Bottom of page 5: What do you mean that continous policies can't be parameterized? Aren't most policy gradient algorithms continous with parameterized policies?

Is the no-op action required to make BC fail?

Why only ensembles of 2? If you have 3 what happens in the grid env?

The authors mention that given an expressive enough policy, it should be possible to imitate any policy and thus the worst-case experts cannot prevent cloning. I would argue that a stronger representation such as a deep network would make the problem easier since deep networks are very susceptible to adversarial attacks and will likely over fit and poorly generalize given finite amounts of demonstrations.

[1] Brown et al. "Machine teaching for inverse reinforcement learning: Algorithms and applications."
[2] Yuzhe et al. "Policy poisoning in batch reinforcement learning and control."


**Experience Assessment:**

I have published in this field for several years.

**Review Assessment: Checking Correctness Of Derivations And Theory:**

I assessed the sensibility of the derivations and theory.

**Review Assessment: Checking Correctness Of Experiments:**

I carefully checked the experiments.

**Review Assessment: Thoroughness In Paper Reading:**

I read the paper at least twice and used my best judgement in assessing the paper.

---

> ### Author Response · Authors · 2019-11-15
> **Response**
>
> We would like to thank AnonReviewer3 for their comments, and their excitement in possible future directions that may stem from our work.
>
> 1. We have fixed many typos, and made other changes to make the paper easier to follow / read.
> 2. While more experiments may enhance our work, we stand by that the current experiment is enough to show validation in our concept and idea. Our environment is the most straightforward environment to demonstrate our “unclonableness” concept, which we elaborate in the experiment section.
> 3. Second paragraph in 2.3 Policy Ensembles has been removed. As well, we have shifted away from the privacy of human demonstrations. We would like to thank AnonReviewer3 for the suggestion on this motivation. We have made changes to the introduction, to focus on inputs generated adversarially, as well as situate it in the current RL space of mainly adversarial attacks on learning policies (whereas ours can be considered a defense against imitation learning).
> 4. We thank AnonReviewer3 for the paper suggestions, and have incorporated them into the new Related Work paragraph, where we situate our work within the field of Adversarial and Private Reinforcement Learning.
> 5. Min-max approaches generally come from min-maxing the same quantity. In our case, we are minimizing and maximizing different objectives. However, scaling using GANs may be potential areas of future work.
> 6. Equations 4,5 had descriptions / explanations that were in the caption of a figure, have now been moved to be right after introducing the equations.
> 7. Bottom of page 5: Currently, popular continuous policies are gaussians or mixture of gaussians. However, we suspect that it may take more expressive policies (that are not gaussian / mixture of gaussian) to fully take advantage of our method in continuous action space. Otherwise, exploitations of how the cloner clones could occur, and would not lead to interesting examples. Consider the classic CartPole Swing-up. If the BC, as usual, is performed by minimizing L-2 loss, then simply having 2 experts, one which moves right, and the other which moves left would cause the cloned policy to remain still at the starting point. This would be uninteresting, as it is exploiting our lack of ability to express policies in continuous state space. Similarly, we would like to shy away / put less emphasis from work that exploits NN.
> 8. The Stay action is not necessary, however we found the plots to be more informative (as the agent would run into the wall, which would be equivalent to Stay).
> 9. We noticed that with more contexts, there was more stable learning (less variance across seeds), although that may also be due to hyperparameter changes needed to train more experts. The result / reward difference would also increase with more experts, although not by any significant amount -- n=3 gets reward difference of 30.83, 0.66 std across 3 seeds, up from 27.81 with n=2.
> 10. We are not considering the scenario of finite data and poor generalization, as we instead are considering when the collector has as much data as desired, to create the perfect clone. We show that mathematically and empirically, we can make this perfect clone bad.

---

### Official Review · AnonReviewer1 · 2019-10-23
**Official Blind Review #1**

**Rating:** 1

**Review:**

The paper proposes a method to learn an ensemble of policies that is hard to imitate from their rollout trajectories. I like the idea of introducing the problem of privacy in reinforcement learning, and it is quite essential. However, some concerns are raised after checking the draft, and I believe the paper could be improved if some of the questions are addressed:

* The current experiment could be an interesting demonstrative part to show how the algorithm works. However, there are no robust empirical experiments that the proposed method could achieve comparable performance/accumulated return as the policy ensemble (PE). I think some experiments on popular benchmarks like Mujoco simulation environment, robotics learning tasks, and Atari games are needed to make the point. The paper will become more convincing if the argument is proved on those benchmarks. Also, more experts should be explored (n > 2) in the experiments.

* It is better to have some mathematical/theoretical analysis of the learning behavior of APE. For instance, is there a theoretical guarantee that APE could achieve comparable performance as PE?

* The paper should discuss more details/analysis of the algorithm, like the choice of $\alpha$ and $\beta$, etc., which I think will affect the algorithm a lot.

* Some related literature on privacy in machine learning could be discussed in the related work section.


====Minor that leads to confusion:
-No mention about J and M before Alg 1; It is assumed to be the objective function and environment
-No mention of the hyper-parameter $\alpha$ after equation 2.



**Experience Assessment:**

I have read many papers in this area.

**Review Assessment: Checking Correctness Of Derivations And Theory:**

I assessed the sensibility of the derivations and theory.

**Review Assessment: Checking Correctness Of Experiments:**

I carefully checked the experiments.

**Review Assessment: Thoroughness In Paper Reading:**

I read the paper thoroughly.

---

> ### Author Response · Authors · 2019-11-15
> **Response**
>
> We would like to thank the official blind reviewer for their comments.
>
> In response to:
> “However, there are no robust empirical experiments that the proposed method could achieve comparable performance/accumulated return as the policy ensemble (PE)”
> We assume that the reviewer meant an optimal policy when they refer to PE — in this case, we can choose how optimal the PE trained by PG-APE via ß, how well the APE should perform. Our derivations would lend us to believe that performance of the PE trained through PG-APE should not be of any concern.
>
> We have added literature in the intro and related work section regarding privacy in ML and RL.
>
> Addressing the minor mistakes:
>  $J$ has been labeled in Equations 4,5. $\mathcal{M}$ is the MDP as defined in the Preliminaries
> $\alpha$ was a parameter used in the derivation, and not a hyperparameter for implementations (although it is chosen via $\beta$).

---

### Official Review · AnonReviewer2 · 2019-10-23
**Official Blind Review #2**

**Rating:** 3

**Review:**

Summary: The paper introduces a method for generating trajectories which prevent behavioral cloning in a policy gradient setting by learning varying experts which try to minimize the ability of a cloned policy. It runs experiments on a grid world to validate empirically that cloning is unsuccessful.

Recommendation: While this is a novel concept and interesting, I cannot recommend acceptance in its current state. The paper was a bit hard to follow and I found the experiments not robust enough to fully characterize the method at this point. It is unclear whether this method really would prevent cloning given an apples-to-apples comparison. My understanding from the paper -- which was a bit hard to follow -- is that cloned policies were tabular while the APE policies were NNs. I would be more confident in results if more environment variations were tested, the cloned policies used more current and apples-to-apples comparisons, and overall if there were more clear details about the methodology.

Comments:
+ It might be worth perusing the differential privacy and adversarial attack literature to think about whether demonstrations can simply be noised to retain information while crashing performance. This work seems relevant for example (it was put online in June which is sufficiently before the September deadline to mention it I believe): Behzadan, Vahid, and William Hsu. "Adversarial Exploitation of Policy Imitation." arXiv preprint arXiv:1906.01121 (2019).
+ In the discussion:
"We found in our preliminary results that using an RNN classifier which outputs p(c|τ1:t) simply ended up in with either optimal policies or crippled policies. In both cases, there was a relatively minor difference in performance between the policy ensemble and the cloned policy." --> There are no quantitative results for this so either results should be included and discussed or this should be future work.
+ The algorithm box doesn't really add a whole lot of information other than saying that trajectories are collected and then gradients are updated. It would be really nice to have a very clear picture of what's happening at each point in the algorithm. In its current state the paper is hard to follow and decipher this sequence.  See for example Algorithm one in: https://papers.nips.cc/paper/6391-generative-adversarial-imitation-learning.pdf .
+ Notation-wise, R(t) is a bit unusual notation for the RL literature, the advantage is usually r + \gamma V(s') - V(s), where r+\gamma V(s') is the action value Q(s,a). Given that the advantage is denoted as A(s,a), it would be clearer I think to use the Q(s,a) notation. Also the notation changes from section 2.2 to section 3.2 from A(s,a) to A(t). Keeping consistent notation would make this paper a lot easier to read.
+ The related work section is in the middle of the paper. it'd be nice to have it earlier to set the context of the work.
+ In the multiple policies section, a recent work has shown how to learn multiple policies from multiple experts using a mixture of experts framework -- though they frame it as options: Henderson, Peter, Wei-Di Chang, Pierre-Luc Bacon, David Meger, Joelle Pineau, and Doina Precup. "Optiongan: Learning joint reward-policy options using generative adversarial inverse reinforcement learning." In Thirty-Second AAAI Conference on Artificial Intelligence. 2018.
+ Part of the way this defeats behaviour cloning is through the assumption that there are multiple trajectories to be learned from. It would be interesting to see if methods like the one above or any of the others mentioned can recover optimal performance from noisy trajectories by similarly learning multiple policies. In fact,
+ "Policy Gradient (PG) (Sutton et al., 2000) and its variants (Schulman et al., 2015) aim to directly learn
the optimal policy π, parameterized by θ." --> I think some other citations of variants should be added for the final version instead of only referencing Schulman 2015. There are a lot now, so maybe adding PPO, DDPG, and a few others might be nice. Otherwise you could also just cut out the variants bit since it's not necessary.
+ All first quotation marks are backwards in the document
+ I think the experiments ran were a bit lacking in robustness and details. Since this is an adversarial method, I would expect more variance across seeds and 3 seeds may not be enough to characterize this. Table 1 has +/- but does not state what this represents. Standard Deviation or Standard error? Does Table 1 represent returns for rolled out policies after learning or across all episode returns during learning? For the behavioural cloning method, it says a "tabular policy" was trained. Does this mean that the experts were trained using policy gradients and neural networks while the behavioural cloning method used a tabular policy? If so, I think this would be at a detriment to the method being tricked. I think it is a necessary condition to validate this method across several gridworld environment variations, seeds, and with more robust cloning methods (if in fact the behavioural policy was underpowered (tabular vs. nn). Overall, it would be great to have more details. While the visualizations of the gridworld itself were nice, I think they took up a lot of space which could be replaced with more detailed explanations and robust quantitative results.


**Experience Assessment:**

I have published one or two papers in this area.

**Review Assessment: Checking Correctness Of Derivations And Theory:**

I assessed the sensibility of the derivations and theory.

**Review Assessment: Checking Correctness Of Experiments:**

I carefully checked the experiments.

**Review Assessment: Thoroughness In Paper Reading:**

I read the paper at least twice and used my best judgement in assessing the paper.

---

> ### Author Response · Authors · 2019-11-15
> **Response**
>
> We would like to thank AnonReviewer2 for their insightful comments.
>
> Specifically, we targeted to address your two concerns. (1) To make the paper easier to follow, we situate the paper in the current RL literature and explain our approach in the introduction, as well as improving the overall language. (2) To clarify methodology, we have rephrased, reordered, the Method and Experiments section. However, we would like to note that our Appendix should answer any questions about practical implementation.
>
> 1. We thank AnonReviewer2 for their paper suggestions. We have included them accordingly. As well, we included a new section to properly place our work with respect to other adversarial RL work. While the idea of noising demonstrations may seem relevant, we are actually completely separate, as we assume that the observer can clone based off of perfect observations. Our novel approach focuses on protecting the policies, rather than an explicit attack any learners.
> 2. As this section is a discussion and future work, we have written what we speculate to be exciting avenues. Specifically, if our method cannot train bad ensembles in a particular environment, that would imply that behaviour cloning should excel in cloning expert demonstrations of that environment. The notion of using cloning sequential policies is not new, in fact is very successful [1], which is why we feel that it is important to include how to tackle such types of cloning. Similarly, the paper AnonReviewer2 referenced above wrote the future work and outlined a potential algorithm.
> 3. We have added a bit more detail in the Algorithm box, and changed the figures to be more illustrative of the algorithm. As well, we have made changes to accommodate the requests.
> We have “corrected” the notation for $A(s_t, a_t)$ in the preliminaries -- we clarify that $R_t$ and $A_t$, wich appear frequently in PG papers such as GAE and PPO and papers that vary PG such as Strouse et al [2], are the sample estimates. We additionally would like to differentiate between the returns from trajectories collected under different policies, namely $\pi_{c^{(i)}}$, and $\pi_o$, which is why we superscript the policy as is also done in other papers. Based on our revisions, our notation follows the current conventions, and is consistent throughout the paper.
> 4. Related work has been moved, to better situate the paper within the field of adversarial RL and confidential ML. As well, we would like to thank AnonReviewer2 for their suggestions on additional papers to include.
> 5. “...multiple trajectories to learn from… In fact,” We are unsure if the reviewer has forgotten to write something, or if there was a typo. However, we did notice, with our other preliminary experiments, that cloning 2 tabular policies, with an RNN to predict which policy to use, was effective in combatting our proposed strategy. This experiment is why we mention in our Future Work section that strong representations should and can clone no matter what.
> 6. “Otherwise you could also just cut out the variants bit since it's not necessary. “
> The original intention was to cite the different possible ways to estimate the gradient used in PG, which is nicely summarized in GAE. We have changed the section to clarify this.
> 7. While we take into the account of the suggestion to remove the gridworld visualizations, however we feel that it is quite instructive to see with the colours how the two experts learn to sacrifice “screw over” the observer.
> 8. “Does Table 1 represent returns for rolled out policies after learning or across all episode returns during learning” The description of Table 1 has been updated to address this. It was mentioned prior that a reasoning for discrete state space was for the closed form solution of the expected returns from each state, which is how the returns are calculated. As well, the caption is more descriptive of what it contains. Most noteworthy is that we added 2 more seeds, which did not affect the statistics very much...
> 9. The only possible “overpowering” would be how the cloner can only have 1 table, while we have $n$ tables. However, this would be analogous to real-life cloning humans, as there are many humans, and only one parameterization of the cloned policy.
>
> [1] Rahmatizadeh, Rouhollah. "Learning robotic manipulation from user demonstrations." (2017).
> [2] Strouse, D. J., et al. "Learning to share and hide intentions using information regularization." Advances in Neural Information Processing Systems. 2018.

---

### Author Response · Authors · 2019-11-15
**Summary of Changes**

1. To better situate this in other “adversarial” methods in RL, we changed the Introduction to create the distinction between current methods, which serve as attacks to the RL algorithms, and ours, which is purely a protection method to prevent BC.
We included a new section in Related Work to reflect this change, as well as moved Related Work section to section 3 (right after preliminaries).
2. Edited the methodology to make it more clear — these were all tabular representations.
3. Notation fixes — preliminaries were edited to be consistent with remaining notation. We make it clear we have similar notation to that of other policy gradient papers.
4. Figure 1 changed to incorporate both figure 1 and 2 from previous iteration to improve method clarity.
5. Figure 2 caption moved to right after the Equation 4,5
6. Equation 5 broken to two pieces, also to add clarity
7. Table 1: added more seeds for PG-APE, as well as more explanation for how the numbers were collected, and what they represent.

---

### Decision · Program_Chairs · 2019-12-19

**Decision:**

Reject

**Comment:**

Although the reviewers appreciated the novelty of this work, they unanimously recommended rejection.  The current version of the paper exhibits weak presentation quality and lacks sufficient technical depth.  The experimental evaluation was not found to be sufficiently convincing by any of the reviewers.  The submitted comments should help the authors improve their paper.